# Thrombin Activity in Rodent and Human Skin: Modified by Inflammation and Correlates with Innervation

**DOI:** 10.3390/biomedicines10061461

**Published:** 2022-06-20

**Authors:** Valery Golderman, Shani Berkowitz, Shani Guly Gofrit, Orna Gera, Shay Anat Aharoni, Daniela Noa Zohar, Daria Keren, Amir Dori, Joab Chapman, Efrat Shavit-Stein

**Affiliations:** 1Department of Neurology, The Chaim Sheba Medical Center, Ramat Gan 52626202, Israel; valery.rodionov@gmail.com (V.G.); shanihberkowitz@gmail.com (S.B.); shanygo@gmail.com (S.G.G.); geragra@post.tau.ac.il (O.G.); aharonianat@gmail.com (S.A.A.); zohardanielle@gmail.com (D.N.Z.); amir.dori@gmail.com (A.D.); dariaker9@gmail.com (D.K.); joabchapman@gmail.com (J.C.); 2Department of Neurology and Neurosurgery, Sackler Faculty of Medicine, Tel Aviv University, Tel Aviv 6997801, Israel; 3Talpiot Medical Leadership Program, The Chaim Sheba Medical Center, Ramat Gan 52626202, Israel; 4Department of Physiology and Pharmacology, Sackler Faculty of Medicine, Tel Aviv University, Tel Aviv 6997801, Israel; 5Robert and Martha Harden Mental and Neurological Diseases, Sackler Faculty of Medicine, Tel Aviv University, Tel Aviv 6997801, Israel; 6The TELEM Rubin Excellence in Biomedical Research Program, The Chaim Sheba Medical Center, Ramat Gan 52621, Israel

**Keywords:** thrombin activity, skin, small-fiber neuropathy

## Abstract

Thrombin is present in peripheral nerves and is involved in the pathogenesis of neuropathy. We evaluated thrombin activity in skin punch biopsies taken from the paws of male mice and rats and from the legs of patients with suspected small-fiber neuropathy (SFN). In mice, inflammation was induced focally by subcutaneous adjuvant injection to one paw and systemically by intraperitoneal lipopolysaccharides (LPS) administration. One day following injection, thrombin activity increased in the skin of the injected compared with the contralateral and non-injected control paws (*p* = 0.0009). One week following injection, thrombin increased in both injected and contralateral paws compared with the controls (*p* = 0.026), coupled with increased heat-sensitivity (*p* = 0.009). Thrombin activity in the footpad skin was significantly increased one week after systemic administration of LPS compared with the controls (*p* = 0.023). This was not accompanied by increased heat sensitivity. In human skin, a correlation was found between nerve fiber density and thrombin activity. In addition, a lower thrombin activity was measured in patients with evidence of systemic inflammation compared with the controls (*p* = 0.0035). These results support the modification of skin thrombin activity by regional and systemic inflammation as well as a correlation with nerve fiber density. Skin thrombin activity measurments may aid in the diagnosis and treatment of SFN.

## 1. Introduction

Neuroinflammation refers to the activation of an immune response in the nervous system and is commonly associated with neural destruction. The precise mechanisms that drive this process are not fully elucidated, and the coagulation system has been suggested to play a role. Inflammation upregulates the coagulation system in many systemic diseases and can be majorly activated, as seen in disseminated intravascular coagulation (DIC) during severe sepsis [1]. On the other hand, thrombin, a central coagulation factor, affects inflammation by several mechanisms through its cellular protease-activated receptors (PARs), which are localized to various cell types such as endothelial, fibroblasts, smooth muscle, and sensory neurons [2,3].

Thrombin modulates long-term potentiation [4], synaptic transmission, and nerve conduction [5]. Excess thrombin activity is associated with numerous central nervous system (CNS) pathologies, including seizures, induced by lowering the epileptic threshold [4], glioblastoma multiforme edema formation [6,7], and primary inflammatory diseases of the brain such as multiple sclerosis [8]. In the peripheral nervous system (PNS), recent studies demonstrate the role of thrombin in the pathogenesis of diabetic neuropathy [9] and Guillain Barre syndrome (GBS) [10]. In the streptozotocin (STZ) diabetic neuropathy model in rats, and the animal model of GBS, increased thrombin activity in the sciatic nerve was found together with the destruction of the nodes of Ranvier structure, along with nerve conduction impairment [9,10,11]. These findings strongly implicate the cellular mediated thrombin pathway in the process and pathogenesis of neuropathy.

Thrombin has been found to play a significant role via several mechanisms in inflammatory processes of both the CNS and PNS. Thrombin can directly bind endothelial cells, leading to blood–brain barrier (BBB) breakdown; increasing antibodies permeability and lymphocyte infiltration to the CNS; and thus inducing edema, inflammation, and gliosis [12]. Injecting thrombin into peripheral tissues has been shown to induce neurogenic inflammation [13]. Cleavage of the thrombin receptor on sensory nerves stimulates the release of neuropeptides such as substance P, which interacts with neurokinin 1 (NK1R) on endothelial cells to induce extravasation of plasma proteins and edema [13]. Thrombin-activated PAR1 in small diameter nociceptive neurons causes sensitization of the capsaicin receptor transient receptor potential vanilloid subfamily 1 (TRPV1) and promotes the heat-dependent release of the pro-inflammatory neuropeptide calcitonin gene-related peptide (CGRP) [14], thereby suggesting that thrombin and PARs play a role in promoting inflammation and pain in the PNS.

Small-fiber neuropathy (SFN) occurs through various mechanisms including inflammatory processes found in diabetes, GBS, chronic inflammatory demyelinating polyneuropathy, and systemic autoimmune diseases [15]. SFN is a disorder characterized by damage to small, myelinated fibers (Aδ) or unmyelinated C fibers. These somatic fibers carry thermal and pain sensations from the skin to the CNS and their degeneration can lead to multiple symptoms including severe neuropathic pain and thermal allodynia [16]. Additionally, SFN patients are known to suffer from burning pain, shooting pain, allodynia, and hyperesthesia.

Previous studies indicate the presence and function of several coagulation factors in the skin, together with local PAR expression. Thus, it is interesting to study thrombin activity in the skin and whether it is modified locally in pathologies such as neuropathies. Measuring thrombin activity in the skin may serve as a highly accessible marker for the diagnosis and treatment of the coagulation-neuroinflammation process. The aim of our study was therefore to investigate whether thrombin activity can be detected and reliably measured in a skin biopsy and whether different pathological processes modify the activity.

## 2. Materials and Methods

### 2.1. Animals

Adult *Sprague Dawley (SD)* rats (10–12 weeks) and adult *ICR* and *C57BL/6J* mice (8–12 weeks) were housed in standard conditions and fed a standard diet with water available ad libitum. The ambient temperature was set to 22 °C to 23 °C with day/night light control. The protocols of this study were approved by the Sheba Medical Center Committee on the Use and Care of Animals (permit No: 1085-17,1000-15, 1191-19) according to the ARRIVE Guidelines.

For the skin biopsies, mice/rats were anesthetized using Pental (pentobarbital sodium 200 mg/mL); 3 mm punches were obtained from the right and left hind limbs.

### 2.2. Adjuvant-Induced Focal Inflammation

The right hind paw of the *ICR* mouse was injected subcutaneously (S.C.) with 50 µL Adjuvant Complete Freund (BD, 263810) or 50 µL saline. The mice were examined for pain responses using the hot-plate test 1 and 7 days following the injection. Mice were anesthetized using Pental, and skin biopsies for thrombin activity were obtained from the right and left hind paws.

### 2.3. LPS-Induced Systemic Inflammation

The systemic inflammation model in *C57BL/6J* mice was induced by intraperitoneal (I.P.) injection of lipopolysaccharides (LPS) (1 mg/kg, O111:B4, L4130, Sigma, Darmstadt, Germany). One week following injection, the pain response was examined using the hot-plate test. The mice were then anesthetized with Pental, and skin biopsies were obtained from the right hind paw for assessment of thrombin activity.

### 2.4. Hot-Plate Test

Hyper/hypoalgesia was evaluated using the hot-plate test, as previously described [17]. The mice were placed in an acrylic glass cylinder on a heated stage digitally maintained at 51 ± 0.1 °C. The time to heat-response indicated by paw licking, shaking, or jumping was measured. A maximum on-plate time was set to 30 s to prevent skin injury.

### 2.5. Thrombin Activity Measurement

Thrombin activity was assessed using a fluorogenic thrombin substrate (Bachem I-1560, excitation 360 nm; emission 465 nm), as previously described [18]. Human biopsies were thawed and washed 3 times for 5 min in Tris-buffer (contains in mM: 150 NaCl, 1 CaCl_2_, 50 Tris-HCl: pH 8.0) on ice. Human and rodent biopsies were placed in a black 96-well microplate (Nunc, Roskilde, Denmark) containing Tris-buffer. The plate was incubated at 37 °C for 30 min before the addition of the substrate. To eliminate the effect of abundant endopeptidases in the assay, endopeptidase inhibitors (0.1 mg/mL, bestatin hydrochloride—B8385, Sigma; 200 μM prolyl endopeptidase inhibitor II, 537011, Merck Millipore) were added to the substrate. Known bovine thrombin concentrations (T-4648, Sigma) were used to create a calibration curve for each experiment. The specific thrombin inhibitor SIXAC (100 nM, American Peptide Company, Sunnyvale, CA, USA) [19] was added to chosen wells to assess the specificity of the assay. The cleavage of the substrate was measured using a microplate reader (Tecan; infinite 200; Männedorf, Switzerland). Each biopsy was weighed, and the measured thrombin activity was normalized to tissue weight. The results are presented as mU thrombin activity/mg of tissue or relative to the control.

### 2.6. Histology

Zamboni-fixed mouse and rat skin biopsies were sectioned into 16 μm sections and stained with Hematoxylin and Eosin (H&E). Briefly, tissue-containing slides were incubated for 10 s in Hematoxylin (Sigma, HHS-32); washed with water; dipped 3 times in Eosin Y (Sigma, HT110-2-32); washed with water; and dipped in 70% ethanol, 95% ethanol, and 100% ethanol. The slides were viewed using a microscope (Olympus, Tokyo, Japan). The images were analyzed using ImageJ software [20]. The thickness of the epidermis was measured in three independent sections. Epidermal thickness was defined as the distance between the distal border and the dermis–epidermis border.

For small-fiber staining, mouse and rat skin biopsies were fixed using Zamboni, sectioned into 16 μm sections, and adhered to charged slides. Slides were washed with PBS and incubated in a blocking solution (containing: 0.25 M Tris-buffered saline, 5% skim milk powder, 5% normal horse serum, and 0.3% Triton X-100) for 4 h at room temperature. The slides were then incubated with a primary antibody (rabbit anti-PGP9.5, SAB4503057, 1:200 in blocking solution diluted 1:1 in PBS) overnight at 4 °C. On the next day, the slides were washed and incubated with secondary antibody (DyLight488 conjugated donkey anti-rabbit IgG, Jackson ImmunoResearch,1:200 in blocking solution diluted 1:1 in PBS) for 1.5 h at room temperature. Following incubation, the slides were washed, dried, and mounted with an anti-fading mounting medium (DAKO Fluoromount). Sections from each animal were randomly analyzed using an Olympus microscope, and the IENFD, expressed as the average number of nerve fibers per millimeter epidermis, was quantified according to guidelines published by the European Federation of Neurological Societies [21].

### 2.7. Human Patients

Human skin biopsies were obtained following informed consent from 43 adults (age 25–78) who were referred due to pain complaints to determine evidence for skin denervation and assist with the diagnosis of SFN (<5th percentile of nerve fiber density normalized for age). SFN was identified in 14 out of 43 patients, and 19 out of 43 patients were suspected of having an inflammatory cause of neuropathy due to known autoimmune or inflammatory disease or by the presence of clearly elevated inflammatory markers (Appendix A). Patients that were under anti-inflammatory and/or immunosuppression treatment were excluded from the “inflammatory group” and moved to the “non-inflammatory group”.

The study was approved by the Chaim Sheba Medical Center Ethics Committee (6525-19-SMC). This work was reported according to the Strengthening the Reporting of Observational Studies in Epidemiology (STROBE) guidelines.

### 2.8. Skin Biopsy

Skin biopsies (two 3.0 mm punches) were obtained from the distal leg 10 cm above the lateral malleolus. One skin biopsy was fixed in Zamboni fixative and processed using standard procedures to determine the intra-epidermal nerve fiber density (IENFD) [21]. Skin IENFD quantitation was performed by A.D. employing immunofluorescence microscopy with the PGP9.5 pan-neuronal marker (Bio-rad, MCA4750GA). Normal IENFD values were based on published data [22] and adjusted according to controls at the Sheba Medical Center. IENFD below the 5th percentile for age was determined abnormal and consistent with skin denervation, indicating small-fiber sensory polyneuropathy. Another skin biopsy was kept unfixed at −80 °C for 3–8 weeks and then used for thrombin activity analysis.

### 2.9. Statistical Analysis

Unpaired *t*-tests and one-way ANOVA with Tukey post hoc analysis were conducted on normally distributed data. Linear regression was conducted for thrombin activity versus age and thrombin activity versus IENFD percentile. The results are expressed as the mean ± SEM, and *p*-values less than 0.05 were considered significant. All statistical analyses were conducted using Prism GraphPad (version 8.0.1 for Windows, GraphPad Software, La Jolla, CA, USA).

## 3. Results

### 3.1. Mouse versus Rat Skin

Thrombin activity was measured in the glabrous skin of rats and mice using an enzymatic specific fluorescent method. Skin biopsy was placed in a 96-well black plate, and the plate was incubated at 37 °C for 30 min to allow the thrombin to defuse out of the tissue. Following the incubation, a thrombin-specific substrate was added, and fluorescence intensity was measured for 50 min (Figure 1).

Both mice and rats showed detectable and specific thrombin activity in their hind paw skin, which was essentially abolished in the presence of SIXAC, a specific thrombin inhibitor (Figure 2A,B). In addition, we found that the rat skin biopsies weighed significantly more compared with the mouse skin (6.4 ± 0.25 and 4.7 ± 0.17 mg, respectively, *p* < 0.0001, Figure 2C). The measured thrombin activity was therefore normalized to the biopsy weight. Accordingly, significantly higher activity was measured in rats’ skin compared with in the mice skin (2.5 ± 0.46 and 0.7 ± 0.13 mU/mg, respectively, *p* = 0.0004, Figure 2D).

H&E skin staining revealed a significantly thicker epidermis layer in rats compared with in mice (97.23 ± 2.6 and 53.9 ± 2.3 µm, respectively, *p* < 0.0001, Figure 3A,B). In addition, staining of the small-fibers showed significantly higher fiber density in the rat skin compared with in the mouse skin (22.5 ± 1.2 and 12 ± 0.97, respectively, *p* < 0.0001, Figure 3C,D).

### 3.2. Skin Thrombin Activity in Adjuvant-Induced Focal Inflammation

To examine the effect of focal inflammation on thrombin activity in the skin, we subcutaneously injected the adjuvant into the hind paw of *ICR* mice. After 24 h, adjuvant injected mice showed slightly higher heat sensitivity, which did not reach significance (Figure 4A). One week following the injection, the mice showed significantly higher heat sensitivity compared with the control (14.9 ± 2 and 25 ± 1.9 s, respectively, *p* = 0.009, Figure 4A).

We measured thrombin activity both in the skin of the adjuvant-injected and the contralateral paws as well as from the control mice. After 24 h, thrombin activity was higher in the skin of the adjuvant-injected paw, compared with that of the contralateral and control paws (3.7 ± 0.44, 0.82 ± 0.11 and 1 ± 0.22, respectively, *p* = 0.0009, Figure 4B). One week following adjuvant injection, thrombin activity in the skin of the injected paw declined, but both paws, i.e., injected and contralateral, showed significantly increased activity compared with the controls (2.7 ± 0.53, 2.3 ± 0.27, and 1 ± 0.24, respectively, *p* = 0.026, Figure 4B).

### 3.3. Skin Thrombin Activity in LPS-Induced Systemic Inflammation

To examine the effect of systemic inflammation on thrombin activity in the skin, we induced inflammation in the *C57BL/6J* mice using LPS injection. In contrast to focal inflammation, LPS mice did not show increased heat sensitivity one week following the induction compared with the controls (18.6 ± 1.89 and 18.6 ± 1.7 s, respectively, *p* > 0.99, Figure 4C). However, after one week, LPS-injected mice showed significantly higher thrombin activity in the skin compared with the controls (1.9 ± 0.24 and 1 ± 0.24, respectively, *p* = 0.023, Figure 4D). Nevertheless, this level of activity was not as high as the thrombin activity detected following focal inflammation.

### 3.4. Thrombin Activity in Human Skin

We extended our findings by measuring thrombin activity in human skin biopsies from patients with suspected SFN. The biopsy was taken from the leg, 10 cm above the lateral malleolus of the patient, classified as non-glabrous hairy skin. The skin was stored at −80 °C and analyzed after thawing. Some human skin biopsies contained blood; therefore, all biopsies were washed in a cold buffer before analysis (Figure 5).

Patients were classified according to small-fiber density counts, determined as SFN patients when the small-fiber density was severely reduced, i.e., indicating skin denervation (<5% percentile normalized per age, Figure 6A). In addition, we classified patients as “inflammatory” according to known autoimmune inflammatory disease or the presence of inflammatory markers (Appendix A). Those without evidence of inflammation were classified as idiopathic. We found no correlation between thrombin activity and age when we analyzed measurements from all patients (*p* = 0.2034, Figure 6B). We found a significant correlation between thrombin activity and the percentile of IENFD in all studied patients (*p* = 0.0356, Figure 6C). However, when skin showed severe denervation (SFN), no correlation between thrombin activity and percentile was seen (*p* = 0.6491, Figure 6D). Patients with inflammatory disorders/markers showed lower thrombin activity in the skin compared with idiopathic patients (0.595 ± 0.09 and 1.08 ± 0.12 mU/mg, respectively, *p* = 0.0035, Figure 6E).

## 4. Discussion

In this study, we demonstrated intrinsic thrombin activity in the skin of rodents and humans for the first time. This activity was found to be correlated with the IENFD and modified by inflammation. We adapted our well-established method for thrombin activity measurement [18,23,24] by adding an incubation period to allow for the release of thrombin from the skin. We first demonstrated measurable thrombin activity in 3 mm biopsies from the glabrous skin of mice and rats. The specificity of the method was verified by adding SIXAC, a potent and specific thrombin inhibitor [19] that completely blocked the measurable thrombin activity. We found that rats have significantly higher thrombin activity in their skin in comparison with mice. H&E staining revealed that rats have a significantly thicker epidermis layer compared with mice, which is consistent with previous studies [25]. Thrombin activity in the skin may derive extrinsically from the blood or intrinsically from structures such as peripheral nerve and glial cells. Since in our hands, both mouse and rat biopsies contained undetectable levels of blood, we assume that the measured thrombin is mostly locally derived and released during the incubation period. In previously studied tissues, thrombin activity is locally produced by various neurons surrounding glia cells and its levels were modified in diseased states [24,26]. Analogously, in the skin, we assumed that thrombin activity is produced by neurons, glial cells, or local skin cells, which affects neural function. Both the dermis and epidermis contain nerve fibers that innervate the skin and its associated structures, such as sweat glands, hair follicles, and Meissner and Pacinian corpuscles [27]. In addition, we showed significantly higher IENFD in rat skin compared with in mouse skin, which is consistent with previous studies [28,29]. This may well explain the higher thrombin activity we found in the rat skin. The correlation between IENFD and thrombin activity was also currently demonstrated in normal human skin samples. It is thus suggested that in normal skin samples, the basal thrombin activity is correlated with small-fiber density and may serve as an alternative biomarker for relative denervation.

The link between inflammation and neuropathy is well established [30]; therefore, we tested thrombin activity using our novel method in two mice models of adjuvant-induced focal and LPS-induced systemic inflammation. We hypothesized that both focal and systemic inflammation cause peripheral neuronal damage, including to the small-fibers in the skin and, therefore, affect thrombin activity levels in the skin. We evaluated the functional neuronal damage using the hot-plate test [17]. In the adjuvant-induced model, we found significantly increased heat sensitivity one week, but not one day, following adjuvant injection. In contrast, significantly elevated thrombin activity was measured one-day post-adjuvant injection. followed by a significant but more modest increase one-week post-adjuvant injection. Moreover, one week following the adjuvant injection, the increase in thrombin activity was detected both in the injected foot skin and in the contralateral foot, which indicates that the focal inflammation spreads and becomes systemic over time, an interpretation supported by the increase in thrombin activity found in the systemic inflammatory LPS model. In contrast to the findings in the adjuvant focal inflammation model, we found no changes in heat sensitivity in the LPS model. Despite this, we were able to measure significantly increased thrombin activity in the skin. This important observation in both models points out the high sensitivity of this novel method to detect pathological processes before they can be assessed by behavioral tests. It is also important to note that the increased thrombin activity in the skin of the adjuvant-injected mice model (focal inflammation) was more significant than the increase in the LPS mice model (systemic inflammation). This suggests a correlation between the increase in thrombin activity and the level of the damage. To eliminate the possibility that the variance between the models is due to the strains of mice we used, we examined basal thrombin activity in these strains and found no significant difference (Appendix A). Further experiments are needed to determine the source of the excess thrombin activity. This increased activity can be due to physical damage to neurons and glia, glial activation, or increased neural activity as a compensatory feedback mechanism for fiber loss in the skin [31]. The role of high thrombin levels in neuropathy is supported by studies demonstrating that the inhibition of thrombin in inflammatory diseases protects from neural deficits [10]. We have found that nerve crush leads to increased intrinsic thrombin activity in the nerve [24]. The results of the present study extend the association between thrombin activity and small-fiber neuropathy in the skin.

Human skin biopsies are obtained routinely for the diagnosis of SFN. Therefore, we extended our study by testing thrombin activity in human skin biopsies from patients with suspected SFN. It is complex to directly compare the human and rodent skin thrombin levels since the human samples were first frozen and subsequently thawed. Based on our previous study with frozen tissue [32], we can assume that the measurable activity is lower compared with the activity in fresh tissue. We analyzed the data from patients and found a significant correlation between nerve-fiber density and thrombin activity. We hypothesize that in normal conditions (above the 5th percentile), thrombin is secreted from neurons or the surrounding glia and is a marker for the normal activity of the fibers. Decreased fiber density is probably associated with both fewer nerve fibers and glia accompanied by lower thrombin activity. It is reasonable to speculate that the main factor determining thrombin activity is the density of nerve fibers innervating the skin. The exact levels of skin thrombin activity may also be determined by local disease processes such as inflammation. Patients with inflammatory conditions have significantly lower thrombin activity in the skin compared with idiopathic patients. It is interesting to note the opposite effect of inflammation on thrombin activity in the skin of mice (increased activity) compared with humans (decreased activity). In well-defined inflammatory models conducted in mice, we examined the effects in a relatively short period following inflammation induction (up to one week). In contrast, in humans, the time period consists of different progression stages. Patients’ skin biopsies are usually collected weeks, months, and even years after inflammation occurs. Therefore, we assume that an initial increase in thrombin activity occurs at the start of inflammation (as measured in mice skin), which is then followed by skin molecular and structural changes causing decreased thrombin activity (as measured in human skin). In addition, it is important to note that the skin we examined in rodents is glabrous and that the human skin of patients is hairy. It is well known that these two types of skin have different structures [33] and possibly different thrombin activity patterns. Furthermore, rodent punches are relatively clean and contain mostly skin layers and, in some cases, muscle fibers. In contrast, the punches obtained from humans sometimes had blood and a significant fat layer. This adds additional variation between patients and may add further complexity to the results. Furthermore, our data indicate a low but significant correlation between thrombin activity and IENFD percentile in the inflammatory group but not in the idiopathic group. We suggest that these groups differ in their clinical features, with the inflammatory group being better defined while the idiopathic group has patients with a variety of medical issues in different stages. In the inflammatory group at this stage of disease, low IENFD correlates with low thrombin activity. In the idiopathic group, there are a number of severely affected patients with high thrombin activity, possibly indicating an ongoing pathogenic process. The low correlation coefficient may be due to additional important factors affecting the neural integrity in the skin such as the specific pathogenesis, time course of the disease, regeneration, and treatments.

In conclusion, we have established a new specific method for measuring thrombin activity in rodent and human skin. We found that this method is sensitive enough to detect changes in thrombin activity in systemic and focal inflammation. Furthermore, changes in skin thrombin activity following inflammation precede changes in symptoms, such as heat sensitivity. In addition, we demonstrate a significant correlation between human skin thrombin activity and small-fiber density. Furthermore, we found decreased thrombin activity in the skin of patients with inflammatory conditions compared with that of idiopathic patients. Skin thrombin activity is therefore a useful novel marker linked to both inflammation and nerve fiber density. The elevated skin thrombin activity levels also suggest thrombin as a marker and therapeutic target in inflammatory neuropathies.

## Figures and Tables

**Figure 1 biomedicines-10-01461-f001:**
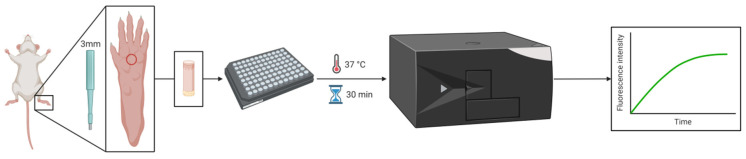
Thrombin activity in rodent skin-method flow diagram: for measuring thrombin activity in rodent skin, a 3 mm punch was obtained from the hind paw, placed in a 96-well black plate, and incubated at 37 °C for 30 min. Following the incubation, a specific thrombin substrate was added, and the fluorescence intensity over time was measured. Created with BioRender.com (accessed on 25 November 2021).

**Figure 2 biomedicines-10-01461-f002:**
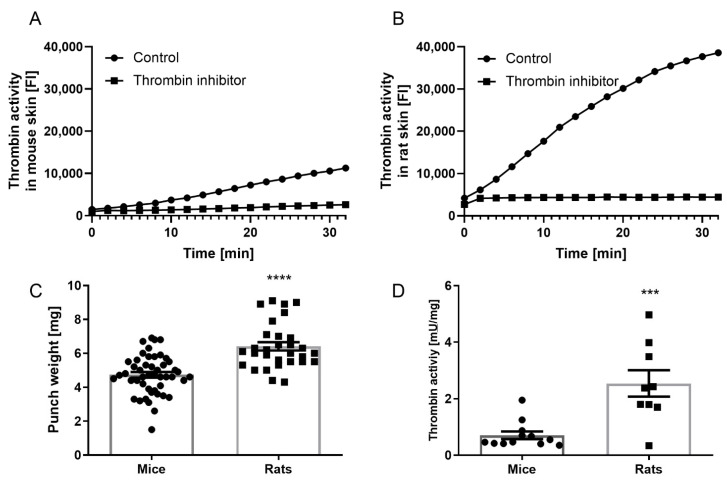
Thrombin activity in healthy mice and rats: representative graphs of measurable thrombin activity that was detected in mice (**A**) and rat (**B**) skin. This activity was fully inhibited by a specific thrombin inhibitor. (**C**) Biopsy weight was significantly lower in mice (n = 30) compared with in rats (n = 48). (**D**) Normalized thrombin activity was significantly higher in rats’ skin (n = 9) compared with in the mice skin (n = 12). Results are presented as mean ± SEM, *** *p* = 0.0004, **** *p* < 0.0001. Scale bar—100 µm.

**Figure 3 biomedicines-10-01461-f003:**
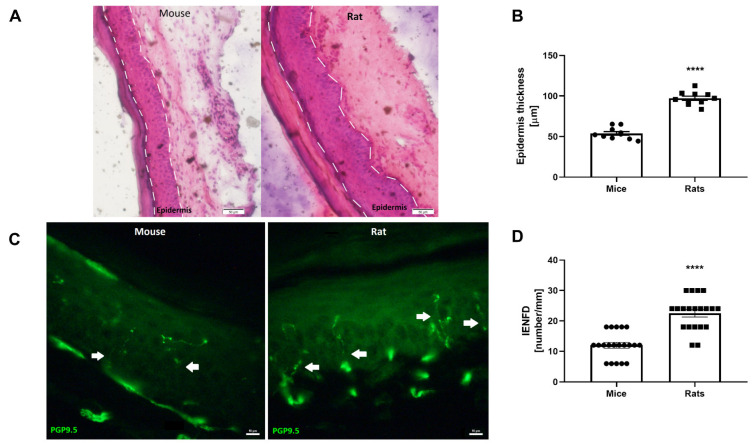
Structural differences between mouse and rat skin: (**A**) representative H&E staining of mouse and rat skin. Scale bar—50µm. White dashed lines represent the borders of the epidermis. Epidermis thickness was defined as the distance between the borders. (**B**) The epidermis layer was significantly thicker in rat skin compared with in mice skin, n = 10. (**C**) Representative PGP9.5 staining of mouse and rat skin. Scale bar—10µm. White arrowheads mark small-fibers. (**D**) The fiber density was significantly higher in rat skin compared with in mouse skin, n = 20. Results are presented as mean ± SEM, **** *p* < 0.0001.

**Figure 4 biomedicines-10-01461-f004:**
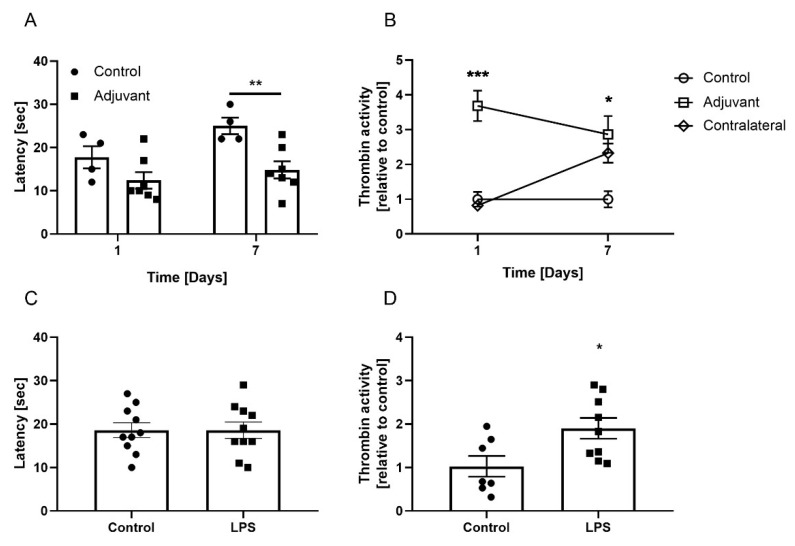
Heat sensitivity and thrombin activity in the skin of focal and systemic inflammation-induced mice. Adjuvant-induced focal inflammation in *ICR* mice: (**A**) Significantly higher heat sensitivity was found one week following adjuvant injection compared with the controls. n = 10 and 4, respectively. (**B**) Thrombin activity in the injected paw was significantly higher compared with the contralateral paw and the control group 24 h following adjuvant injection. Thrombin activity in the skin of the injected and the contralateral paws was significantly higher compared with the control group one week following adjuvant injection. n = 7 and 8. LPS-induced inflammation in *C57BL/6J* mice: (**C**) No difference in heat sensitivity between the groups was found one week following LPS injection. n = 10. (**D**) Significantly higher thrombin activity was found in the skin of LPS mice compared with that of the controls. n = 9 and 7, respectively. Thrombin activity results are presented relative to the controls. Results are presented as mean ± SEM, * *p* < 0.026, ** *p* < 0.009, *** *p* < 0.0009.

**Figure 5 biomedicines-10-01461-f005:**
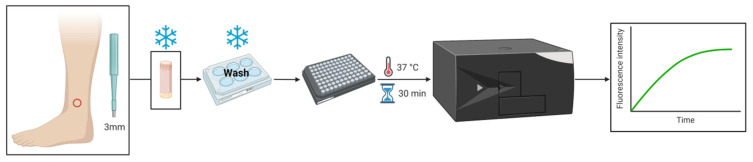
Thrombin activity in human skin—method flow diagram: For measuring thrombin activity in human skin, a 3 mm punch was obtained from the leg (10 cm above the lateral malleolus) and transferred to −80 °C until the thrombin activity assay. The biopsy was thawed and washed 3 times for 5 min on ice. Following this, it was placed in a 96-well black plate and incubated at 37 °C for 30 min. After incubation, a specific thrombin substrate was added and the fluorescence intensity over time was measured. Created with BioRender.com (accessed on 25 November 2021).

**Figure 6 biomedicines-10-01461-f006:**
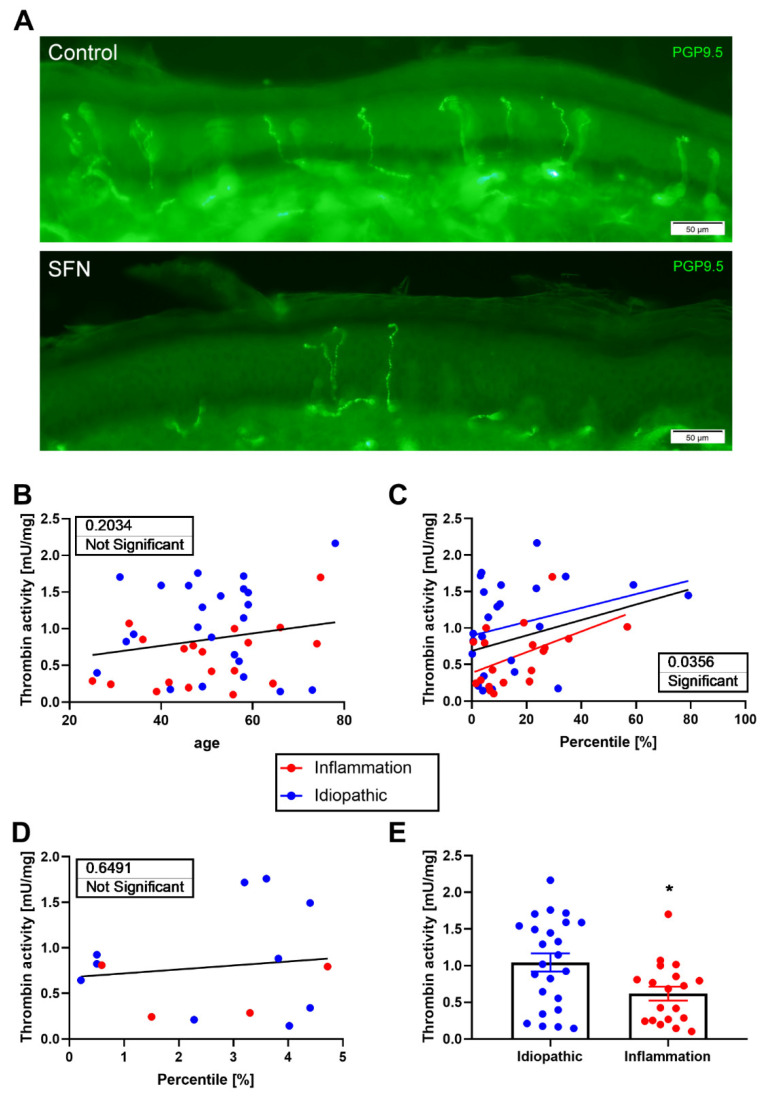
Thrombin activity in human skin: (**A**) Representative pictures of small-fiber staining of control (upper panel) and SFN (lower panel) patients. (**B**) No correlation was found between thrombin activity and age in all patients. *p* = 0.2, R^2^ = 0.04, n = 43. (**C**) A significant correlation was found between thrombin activity and percentile in all patients. *p* = 0.035, R^2^ = 0.1 for all patients; *p* = 0.032, R^2^ = 0.24 for inflammation; and *p* = 0.15, R^2^ = 0.09 for idiopathic. n = 43. (**D**) No correlation was found between thrombin activity and percentile in SFN patients. *p* = 0.6, R^2^ = 0.02, n = 14. (**E**) Significantly increased thrombin activity was found in idiopathic patients (blue) compared with in inflammatory patients (red). n = 23 and 20, respectively. Results are presented as mean ± SEM, * *p* = 0.0035.

## Data Availability

All data are attached in the “Non-published Material”.

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
