# Peer review of "Thrombin Activity in Rodent and Human Skin: Modified by Inflammation and Correlates with Innervation"

_biomedicines, 2022, doi:10.3390/biomedicines10061461_

Round 1

Reviewer 1 Report

In this revised manuscript, the authors answered most of the previous questions and comments properly. However, there are still some remaining questions in the revised MS:

.Figure 3C and Fig 6A: these images are blurry under epifluorescent microscopy. Using confocal images instead is strongly recommended.

. Figure 4B. The figure legends are to small to comprehend. Please make necessary adjustment (enlargement or different colors).

. Figure 6.

-Figure legend in Page 10, Line 277. Figure 1 should be corrected as Figure 6.

-Figure 6B-D. What are the respective correlation coefficients for these figures? If the inflammatory and idiopathic groups are calculated as separate groups, what are the coefficients? Will you get significant difference for idiopathic groups in Fig B-D?

Author Response

Dear Reviewers,

Thank you for your thorough and positive review of the manuscript titled “Thrombin Activity in Rodent and Human Skin: Modified by Inflammation and Correlates with Innervation” (#1760817). We appreciate your positive feedback and have thoroughly re-reviewed the manuscript. We have revised the manuscript according to the reviewers’ comments. Below is our point-by-point response to each reviewer in blue. Our changes in the text are written in blue.

We hope that all issues have been adequately addressed and we have edited the manuscript accordingly.

We hope that the current form of the article meets the high standards of the “Biomedicines” journal.

Thank you for this opportunity and your support,

Efrat Shavit-Stein, on behalf of the authors

Independent Review Report, Reviewer 1

In this revised manuscript, the authors answered most of the previous questions and comments properly. However, there are still some remaining questions in the revised MS:

Figure 3C and Fig 6A: these images are blurry under epi-fluorescent microscopy. Using confocal images instead is strongly recommended.

We agree with the reviewer’s comment that confocal imaging may produce a sharper image. However, the representative figures in Figures 3C and 6A were obtained from our IENFD routine clinical assessment slides and best represent our practice using regular fluorescence microscopy. Confocal microscopy and figure acquisition for IENFD evaluation are time-consuming, thus most of the studies including our own use an epi-fluorescence microscope that is as reliable as confocal [Provitera V, Nolano M, Stancanelli A, Caporaso G, Vitale DF, Santoro L. Intraepidermal nerve fiber analysis using immunofluorescence with and without confocal microscopy. Muscle Nerve. 2015 Apr;51(4):501-4. doi: 10.1002/mus.24338. Epub 2015 Feb 11. PMID: 25043126.].

Figure 4B. The figure legends are too small to comprehend. Please make necessary adjustments (enlargement or different colors).

We thank the reviewer for this important comment. We changed the legends to make them more comprehensible.

Figure 6.

Figure legend in Page 10, Line 277. Figure 1 should be corrected as Figure 6.

Corrected.

-Figure 6B-D. What are the respective correlation coefficients for these figures? If the inflammatory and idiopathic groups are calculated as separate groups, what are the coefficients? Will you get significant difference for idiopathic groups in Fig B-D?

We added the correlation coefficients in figure 6B, C, and D legend for all patients (R2=0.04, 0.035, and 0.02 respectively) and separately for the inflammatory and idiopathic groups in the legend of 6C (R2=0.24 and 0.09 respectively). Indeed, as the reviewer suggested, in Figure 6C there was a significant correlation between thrombin activity and IENFD percentile in the inflammatory groups but not in the idiopathic group (we have added their linear fit curves to Figure 6C). We have commented on this finding in the Discussion: We suggest that these groups differ in their clinical features, the inflammatory group being better defined while the idiopathic group has patients with a variety of medical issues in different stages. In the inflammatory group at this stage of disease, low IENFD correlates with low thrombin activity. In the idiopathic group, there are a number of severely affected patients with high thrombin activity possibly indicating an ongoing pathogenic process. We added this comment to the Discussion (page 11, lines 373-380).

Reviewer 2 Report

After the clarifications in the experiments and in the answers given to the main questions raised, we consider that the article deserves to be published.

Author Response

Dear Editors and Reviewers,

Thank you for your thorough and positive review of the manuscript titled “Thrombin Activity in Rodent and Human Skin: Modified by Inflammation and Correlates with Innervation” (#1760817). We appreciate your positive feedback and have thoroughly re-reviewed the manuscript. We have revised the manuscript according to the reviewers’ comments. Below is our point-by-point response to each reviewer in blue. Our changes in the text are written in blue.

We hope that all issues have been adequately addressed and we have edited the manuscript accordingly.

We hope that the current form of the article meets the high standards of the “Biomedicines” journal.

Thank you for this opportunity and your support,

Efrat Shavit-Stein, on behalf of the authors

Independent Review Report, Reviewer 2

Comments and Suggestions for Authors

After the clarifications in the experiments and in the answers given to the main questions raised, we consider that the article deserves to be published.

We thank the reviewer for the acceptance.

This manuscript is a resubmission of an earlier submission. The following is a list of the peer review reports and author responses from that submission.

Round 1

Reviewer 1 Report

In this manuscript, the authors demonstrated a new model for the measurement of tissue thrombin activities, and attempted to correlate them with IENFD and pain behaviors.  In addition, the authors noted different changes in thrombin activities under local and systemic inflammation, as well as differences across various species as well as chronicity of the trigger. The authors applied morphological, behavioral, and biochemical approaches to augment the significance of thrombin in pain and peripheral neuropathy. As though the approach for measuring human skin thrombin activities is interesting and novel, several key points remain to be explained and clarified:

Major issues:

. What are the advantages of thrombin activities measurement in human skin over skin biopsies for IENFD quantification?

. The authors used two strains of mice for this study (ICR for CFA; C57 for LPS). Please justify the reasons for this study design. Could it be the cause for the differences in CFA and LPS models? Have you observed any differences in thrombin activities in these 2 strains?

. The evidence supporting that mouse IENFD is correlated with mouse thrombin activities is missing in this MS.  

- Fig 2.

.What are the purposes for comparing rat and mice thrombin activities? Also please specify if the mice data was collected from C57 or ICR.  

.Fig 2A and B. Are those data from single animal or pooled?

.Fig 2E and F. The quality of the images are suboptimal. Please replace them with the ones of better quality.

.Fig 2G. Please clarify how the epidermis thickness was defined and measured.

- Fig 3.

.Fig 3B and D. What are the relative thrombin activities? Also the legends for fig 3B is obscure (very hard to read).

. Fig 3C. Do you have the Day 1 behavioral data for LPS?

-Fig 5.

. Fig 5B and C. What is the reasons that the authors used percentiles rather than absolute IENFD numbers? Also it is advised to add in the representative PGP9.5 staining for the SFN and normal groups.

. Page 9 discussion. The authors stated that opposite effects of inflammation on thrombin activities are observed in mice and human, and the time course may be the cause for the difference. Have the authors performed chronic CFA model in mice to see if the effects in human tissues can be recapitulated?

Minor issues:

. There are grammatical and spelling errors throughout the MS. English editing service is advised.

.Page 3 method 2.2. For 50uL CFA injection in ICR mice, please clarify if it is subcutaneous or intraplantar injection.

. Figure 5 is misspelled as Fig 1 in page 7.

. Page 7. SFN is classified as “inflammatory” according to known autoimmune disease or presence of inflammatory markers. Please specify the diseases and the markers for the patients enrolled in the study. Also were the subjects under anti-inflammatory/ immunosuppression agents?

Reviewer 2 Report

It is an interesting work with an original experimental approach to systemic and local thrombin activity, however, I think that additional experiments should be carried out and subsequently a re-evaluation of the text prior to its acceptance.